# Increased respiratory morbidity associated with exposure to a mature volcanic plume from a large Icelandic fissure eruption

Hanne Krage Carlsen [1,2,15✉], Evgenia Ilyinskaya [3,15✉], Peter J. Baxter[4], Anja Schmidt [5,6], Throstur Thorsteinsson [1], Melissa Anne Pfeffer[7], Sara Barsotti[7], Francesca Dominici[8], Ragnhildur Gudrun Finnbjornsdottir [9], Thorsteinn Jóhannsson[9], Thor Aspelund [10], Thorarinn Gislason[10,11], Unnur Valdimarsdóttir [8,12,13], Haraldur Briem[14] & Thorolfur Gudnason[14]

The 2014–15 Holuhraun eruption in Iceland was the largest fissure eruption in over 200 years, emitting prodigious amounts of gas and particulate matter into the troposphere. Reykjavík, the capital area of Iceland (250 km from eruption site) was exposed to air pollution events from advection of (i) a relatively young and chemically primitive volcanic plume with a high sulphur dioxide gas ($SO_2$) to sulphate PM ($SO_4^{2-}$) ratio, and (ii) an older and chemically mature volcanic plume with a low $SO_2/SO_4^{2-}$ ratio. Whereas the advection and air pollution caused by the primitive plume were successfully forecast and forewarned in public advisories, the mature plume was not. Here, we show that exposure to the mature plume is associated with an increase in register-measured health care utilisation for respiratory disease by 23% (95% CI 19.7–27.4%) and for asthma medication dispensing by 19.3% (95% CI 9.6–29.1%). Absence of public advisories is associated with increases in visits to primary care medical doctors and to the hospital emergency department. We recommend that operational response to volcanic air pollution considers both primitive and mature types of plumes.

[1] Environment and Natural Resources, University of Iceland, Reykjavík, Iceland. [2] Section of Occupational and Environmental Medicine, Department of Public Health and Community Medicine, Institute of Medicine, Sahlgrenska Academy at University of Gothenburg, Gothenburg, Sweden. [3] School of Earth and Environment, University of Leeds, Leeds, UK. [4] Cambridge Institute of Public Health, University of Cambridge School of Clinical Medicine, Cambridge, UK. [5] Department of Geography, University of Cambridge, Cambridge, UK. [6] Yusuf Hamied Department of Chemistry, University of Cambridge, Cambridge, UK. [7] Icelandic Meteorological Office, Reykjavík, Iceland. [8] Department of Biostatistics, Harvard T.H. Chan School of Public Health, Boston, MA, USA. [9] The Environment Agency of Iceland, Reykjavík, Iceland. [10] School of Health Sciences, University of Iceland, Reykjavík, Iceland. [11] Landspitali – the National University Hospital, Reykjavík, Iceland. [12] Centre of Public Health Sciences, University of Iceland, Reykjavík, Iceland. [13] Department of Medical Epidemiology and Biostatistics, Karolinska Institutet, Stockholm, Sweden. [14] Chief Epidemiologist, Directorate of Health, Centre for Health Threats and Communicable Diseases, Reykjavík, Iceland. [15] These authors contributed equally: Hanne Krage Carlsen, Evgenia Ilyinskaya. ✉email: hanne.krage.carlsen@amm.gu.se; e.ilyinskaya@leeds.ac.uk

Over 800 million people are estimated to live within 100 km from active volcanoes (2011 data)[1], a distance within which they are potentially exposed to air pollution from volcanic gas and aerosol particulate matter (PM) emissions. Throughout this work, we will refer to volcanic emissions, and unless otherwise stated, our intended meaning is gas and PM emissions, collectively. One of the most hazardous air pollutants commonly found in volcanic emissions is sulphur dioxide ($SO_2$) gas. $SO_2$ has well-defined hourly and daily exposure limits based on results from laboratory studies in volunteers with non-severe asthma[2]. Asthma sufferers are sensitive to low levels of $SO_2$ (<200 ppb) where the reaction is a rapid onset of constriction of the airways. After emission into the atmosphere, $SO_2$ gas transforms into sulphate PM ($SO_4^{2-}$) at a rate that depends on multiple parameters including ambient temperature and humidity, solar flux, and interactions with other PM[3]. There is an undisputed link between a number of negative health outcomes and both acute and chronic exposure to PM from natural and anthropogenic sources (e.g.[4,5]). Size-resolved PM concentration is an important factor impacting on human health, morbidity and mortality, with fine PM being associated with more detrimental effects (e.g.[4]). The evidence base is weak on which, if any, chemical components of the PM mass are individually responsible for the observed negative health outcomes, and there are no established air quality thresholds for individual PM chemical components, such as sulphate.

Some of the most severely polluting volcanic events are basaltic fissure eruptions. Their emissions are typically ash-poor due to the predominantly effusive nature of the eruption. They can release significant amounts of gases (including $SO_2$) and PM into the troposphere, thereby elevating concentrations of air pollutants at ground level[6–13]. The chemical composition of PM in a fissure eruption plume is generally a heterogeneous mixture of sulphate and a multitude of other species, including metal pollutants (e.g. lead, cadmium) (e.g.[10,13,14]). The size is typically very fine, falling within the $PM_1$ or $PM_{2.5}$ size fractions (PM < 1, or <2.5 μm diameter, respectively), e.g.[10,13]. Large fissure eruptions (>1 km³ lava erupted over a few months) in Iceland were included in the UK National Risk Register in 2012[15,16]. This was based on the significant societal consequences across the northern hemisphere caused by the emissions and subsequent air pollution of the Laki eruption in Iceland 1783–1784 CE (14 km³ lava and 120 Mt of $SO_2$ erupted over 8 months)[17,18]. The modelled health impacts of air pollution from Icelandic large fissure eruptions on modern day Europe are also significant[6,19]. However, an opportunity to directly measure the public health impacts of a large fissure eruption did not present itself until Holuhraun 2014–2015 in Iceland (1.5 km³ of lava[20] and 11–19 Mt[8] of $SO_2$ over 6 months). Until then, most of the modern health studies had been based on the prolonged but small fissure eruption of Kīlauea volcano in Hawaii (~4.4 km³ lava erupted between 1983 and up until May 2018[21]). Volcanic air pollution from Kīlauea is locally known as 'vog' (volcanic fog or smog), a collective term for the volcanic gas and PM mixture of variable chemical maturity level. Public health studies have reported increased rates of respiratory symptoms associated with vog exposure, such as acute bronchitis[22–25]. Self-reported symptoms have included a prevalence of cough[24], bronchitis, sinus pain, eye disease and diseases of the circulatory system[23,26]. The longevity of the Kīlauea eruption presents a challenge for public health studies as the same population cannot be compared before and after vog exposure. Comparison is generally made between high- and low-vog exposure zones, but the relatively small study populations and demographic differences between the zones limit the robustness of the conclusions. Kīlauea's activity escalated to a large fissure eruption May–August 2018 (~1.5 km³ lava[21] and 7–14 Mt[27] of $SO_2$ erupted over

3 months); investigations into the health impact of the increased emissions have not been published at the time of writing.

Holuhraun (31 August 2014–27 February 2015), was the largest fissure eruption worldwide since Laki in terms of the erupted volume of lava, gas, and the geographical spread of volcanogenic air pollution in the troposphere. $SO_2$ from Holuhraun caused frequent and severe deteriorations of air quality in Iceland's populated areas despite the remoteness of the eruption site, with the closest towns at ~100 km distance (isolated farms ~70 km). $SO_2$ was monitored and publicly reported in real time by the Environment Agency of Iceland (EAI). The maximum hourly-mean $SO_2$ concentration measured by the EAI network was 2600 μg/m³ in the town of Höfn (at 100 km distance)[28], defined as a 'dangerous' level by the World Health Organisation (WHO)[2]. Peak measurements of up to 20,000 μg/m³ were reported by hand-held sensors operated by the police. In Iceland's capital area, Reykjavík (250 km distance), $SO_2$ concentrations exceeded WHO hourly air quality guidelines of 350 μg/m³ on 34 occasions[10]. $SO_2$ events from 80 μg/m³ to >500 μg/m³ were reported during the first month of the eruption from the UK, Ireland, Austria, Finland, and the Netherlands[7,29]. In an innovation for Iceland, 72-h forecasts of plume advection direction, and the likelihood of $SO_2$ exceeding the WHO air quality guidelines at ground level were produced by the Icelandic Meteorological Office (IMO), which enabled issuing of public advisories and became an important eruption mitigation tool[9,30]. However, after the eruption, Ilyinskaya et al.[10] showed that not all volcanogenic air pollution events may have been successfully predicted by the operational forecasts. The limited size of the spatial computational domain of the forecasting model did not account for a re-entry of the plume once it was advected out of the domain. On several occasions the atmospheric conditions caused the plume to return to Iceland some days later as a chemically mature volcanic cloud in which $SO_2$ had undergone near-complete conversion to sulphate PM (Fig. 1 and Supplementary Movie 1); this was not forecast or included in public advisories. The PM was sampled daily by EAI in Reykjavík for compositional analysis but not analysed in real time. Analysis after the eruption showed that on days impacted by the mature plume, the chemical composition of PM in Reykjavík was altered and dominated by sulphate (60–90% of PM mass[10]). Other components of PM in the mature plume were various trace metals and ammonium[10].

Carlsen et al.[31] previously showed that the episodes of increased $SO_2$ in Reykjavík, caused by advection of a primitive plume, were associated with increases in asthma medication dispensing (AMD), primary care medical doctor (PCMD) visits, and hospital emergency department (HED) visits for asthma and respiratory infections. In this study we assess the register-based health care utilisation after exposure to the non-forecast and non-monitored chemically mature volcanic plume (low $SO_2/SO_4^{2-}$ ratio). We use Reykjavík (~210,000 inhabitants) as the case study location. Reykjavík has low to moderate levels of background anthropogenic pollution[32] and high-quality centralised health care records. To our knowledge, this study provides the most robust evidence to-date that ground-level exposure to a chemically mature volcanic plume leads to increased health care utilisation. The increased heath care utilisation for respiratory disease occurred on the same day or, in some cases, a few days after the exposure. A comparison of the effects of primitive and mature volcanic plumes showed that, although most confidence intervals overlapped, the mature plume appeared more detrimental than the primitive plume for AMD and PCMD for respiratory infections. Our results also indicated that the absence of public advisories for mature plume exposure was associated with increased PCMD and HED visits for respiratory disease in vulnerable groups. For non-respiratory health outcomes, exposure to the

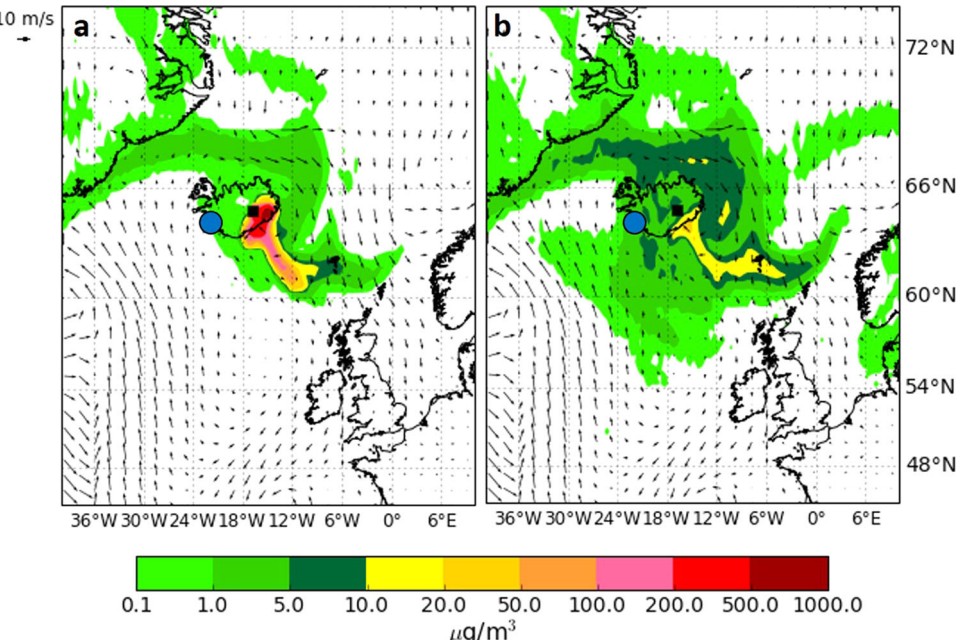

**Fig. 1 Dispersion of primitive and mature plumes.** The ground-level concentrations of (**a**) $SO_2$ and (**b**) $SO_4^{2-}$ (units: $\mu g/m^3$) from Holuhraun eruption, simulated for 20 September 2014. Wind vectors (black arrows) are used to show the plume transport direction. Black square: eruption site. Blue circle: Reykjavík (area approximate). The primitive plume (high mass ratio of $SO_2$ to $SO_4^{2-}$) is dispersed predominantly to the south-east of the eruption site. The mature plume (high mass ratio of $SO_4^{2-}$ to $SO_2$) has returned back from Europe and is inundating the Reykjavík capital area from the south. Figure is modified from Ilyinskaya et al.[10]. See Supplementary Movie 1 for animation.

mature plume was associated with increases in PCMD for circulatory system disease in children. The resulting effect estimates for the total population and vulnerable groups can be used for defining scenarios of health impacts, suggesting mitigation effects, and predicting health care utilisation associated with a similar or larger future fissure eruption in Iceland and elsewhere.

## Results and discussion

**Presence of primitive and mature plumes in Reykjavík.** The presence of the volcanic plume in Reykjavík was identified using air quality monitoring data collected by EAI. Data from two air quality stations in Reykjavík were used in this analysis, Hvaleyrarholt (HVAL) and Grensásvegur (GREN), details in Methods. We considered the plume (of either primitive or mature composition) to be present when the total sulphur in the atmosphere ($S_{total} = S_{gas} + S_{PM}$) averaged over a 24-h interval at HVAL exceeded the maximum 24-h mean $S_{total}$ (30 $\mu g/m^3$) in 9 months before the eruption onset. We found that the plume was present on 33 out of the 123 days of the eruption study period (31 August–31 December 2014). Averaged over the 33 plume-present days, the 24-h mean $SO_2$ concentrations was 76 ± 84 $\mu g/m^3$ (1.5 $\mu g/m^3$ pre-eruption) and $SO_4^{2-}$ was 6.7 ± 4 $\mu g/m^3$ (1.5 $\mu g/m^3$ pre-eruption) as measured at HVAL (Table 1). The maximum 24-hour mean concentration of $SO_4^{2-}$ (15 $\mu g/m^3$) was approximately twice as high during the eruption compared to pre-eruption (6.5 $\mu g/m^3$).

Within the 33 plume-present days, we distinguished between primitive and mature plume compositions based on the $SO_4^{2-}/SO_2$ mass ratio. Primitive plumes, as measured just above the source vent, typically have a $SO_4^{2-}/SO_2$ mass ratio of ~0.01[10,13,14] and this ratio increases with time as the plume ages and chemically matures[10,11]. The chemical maturity level of a volcanic plume is not linearly related to its chronological age, because the rate of maturation is dependent on several factors including the ambient temperature and relative humidity, solar flux, and interactions with other PM. Therefore, in this study, the terms primitive (low ratio of

$SO_4^{2-}$ to $SO_2$) and mature (high ratio of $SO_4^{2-}$ to $SO_2$) denote only the level of chemical maturity and not the chronological age. We identified 21 days of mature plume, defined by the $SO_4^{2-}/SO_2$ mass ratio of ≥0.09, and 12 days of the primitive plume ($SO_4^{2-}/SO_2 <$ 0.09), Table 1. We note that the level of plume maturity is a continuum, and that any ratio selected as the cut-off point is a simplification.

The 24-h $SO_2$ air quality threshold (24-h mean 125 $\mu g/m^3$)[2] was exceeded on 8 out of 12 primitive plume days at HVAL and 10 out of 12 at GREN (Table 1). The threshold was exceeded on 2 out of 21 mature plume days at GREN, with no exceedances at HVAL. For sulphate, evidence-based air quality thresholds do not exist. In a review of anthropogenic pollution in major cities 1930s–1980s, Lioy and Waldman[33] suggested a 24-h mean sulphate threshold of 5 $\mu g/m^3$. In Hawaii, Longo et al.[34] used the 5 $\mu g/m^3$ concentration suggested by Lioy and Waldman[33] as a threshold for identifying volcanic air pollution zones around Kīlauea volcano. On all of the mature plume days, the concentration of sulphate was elevated above the pre-eruption background, and above the 5 $\mu g/m^3$ 'threshold' suggested in the previous studies[33,34] (mean 8 ± 3 $\mu g/m^3$), with only one exception (Table 1).

For $PM_{2.5}$, both mature and primitive plume advection caused a moderate increase compared to the pre-eruption background of 4.8 $\mu g/m^3$,[10] but the overall concentrations still remained low (Table 1). On plume-present days the mean 24-h concentration was ~8 $\mu g/m^3$ for both types of plume, which is considered good air quality. The maximum observed concentration (24-h mean 19-20 $\mu g/m^3$) was below the $PM_{2.5}$ air quality guidelines of 25 $\mu g/m^3$ [2].

On 10 of the 21 mature plume days, the plume's presence in Reykjavík was not forecast by the IMO dispersion model and consequently not included in public advisories (Table 1). These included the 3 days with the highest detected sulphate concentrations (24-h mean ≥14 $\mu g/m^3$). In contrast, public advisories were issued for all primitive plume days. The most

**Table 1 Daily-mean (24 h) ground-level concentrations of sulphur dioxide gas (SO$_2$, measured at Hvaleyrarholt—HVAL, and Grensásvegur—GREN, EAI stations in Reykjavík) and sulphate aerosol (SO$_4^{2-}$, measured at HVAL only) during the eruption study period (31 August–31 December 2014).**

| Date (DD/MM/YYYY) | SO$_2$ 24 h mean, µg/m$^3$ | | PM2.5 24 h mean, µg/m$^3$ (HVAL) | SO$_4^{2-}$ 24 h mean, µg/m$^3$ (HVAL) | SO$_4^{2-}$/SO$_2$ mass ratio (HVAL) | Plume type | Public advisory issued | Outdoor public sports events |
|---|---|---|---|---|---|---|---|---|
| | HVAL | GREN | | | | | | |
| 20/09/2014 | 32 | 15 | 19 | 14 | 0.44 | Mature | No | Yes |
| 06/10/2014 | 49 | 74 | 3.5 | 5.1 | 0.10 | Mature | Yes | |
| 07/10/2014 | 37 | 89 | 4.8 | 1.5 | 0.04 | Primitive | Yes | |
| 08/10/2014 | 62 | 136 | 6.5 | 0.0 | 0.0 | Primitive | Yes | |
| 09/10/2014 | 60 | 108 | 6.6 | 7.9 | 0.13 | Mature | No | Yes |
| 10/10/2014 | 130 | 194 | 7.8 | 9.1 | 0.07 | Primitive | Yes | |
| 11/10/2014 | 15 | 24 | 5.6 | 6.3 | 0.42 | Mature | Yes | Yes |
| 12/10/2014 | 36 | 77 | 7.9 | 8.6 | 0.24 | Mature | Yes | |
| 13/10/2014 | 39 | 50 | 6.6 | 8.6 | 0.22 | Mature | Yes | |
| 14/10/2014 | 110 | 225 | 14 | 14 | 0.13 | Mature | No | |
| 15/10/2014 | 70 | 160 | 16 | 15 | 0.22 | Mature | No | |
| 16/10/2014 | 280 | 341 | 20 | 8.4 | 0.03 | Primitive | Yes | |
| 17/10/2014 | 27 | 45 | 8.8 | 5.8 | 0.22 | Mature | Yes | |
| 18/10/2014 | 30 | 48 | 6.4 | 7.7 | 0.26 | Mature | Yes | |
| 19/10/2014 | 34 | 54 | 6.4 | 8.5 | 0.25 | Mature | Yes | |
| 23/10/2014 | 58 | 124 | 8.2 | 9.5 | 0.16 | Mature | No | |
| 24/10/2014 | 13 | 40 | 5 | 7.6 | 0.57 | Mature | No | |
| 25/10/2014 | 20 | 29 | 4.3 | 6.9 | 0.34 | Mature | Yes | Yes |
| 26/10/2014 | 57 | 106 | 7.7 | 4.8 | 0.09 | Mature | No | |
| 29/10/2014 | 280 | 254 | 8.3 | 7.1 | 0.03 | Primitive | Yes | |
| 30/10/2014 | 130 | 223 | 8.9 | 6.9 | 0.05 | Primitive | Yes | |
| 01/11/2014 | 29 | 50 | 4 | 5.4 | 0.18 | Mature | Yes | Yes (×2) |
| 03/11/2014 | 17 | 22 | 6.6 | 0.0 | 0.0 | Likely primitive | Yes | |
| 04/11/2014 | 320 | 418 | 14 | 10 | 0.03 | Primitive | Yes | |
| 09/11/2014 | 240 | 309 | 10 | 5.5 | 0.02 | Primitive | Yes | |
| 10/11/2014 | 47 | 79 | 9.9 | 3.9 | 0.08 | Likely primitive | Yes | |
| 12/11/2014 | 59 | 105 | 6.2 | 5.5 | 0.09 | Likely mature | Yes | |
| 13/11/2014 | 120 | 196 | 5 | 5.2 | 0.04 | Primitive | Yes | Yes |
| 14/11/2014 | 26 | 53 | 4.2 | 0.89 | 0.03 | Likely primitive | Yes | |
| 17/12/2014 | 18 | 7 | 5.7 | 3.1 | 0.17 | Mature | No | |
| 22/12/2014 | 20 | 28 | 4.5 | 6.7 | 0.35 | Mature | No | Yes |
| 24/12/2014 | 23 | 27 | 4.2 | 5.1 | 0.23 | Mature | Yes | |
| 28/12/2014 | 15 | 28 | 7.4 | 6.7 | 0.45 | Mature | No | |
| Summary | mean ± st dev | | | | | Total | Total advisories | Total sports events |
| Primitive plume | 140 ± 110 | 190 ± 120 | 8.8 ± 4.4 | 5.9 ± 3.1 | 0.032 ± 0.02 | 12 | 12 | 2 |
| Mature plume | 39 ± 24 | 68 ± 54 | 7.5 ± 4.0 | 7.8 ± 3.3 | 0.25 ± 0.1 | 21 | 11 | 7 |

The Table includes days when the volcanic plume from Holuhraun reached Reykjavík. For definition of 'primitive' and 'mature' plume see main text. All units in µg/m$^3$, reported to 2 significant figures. Outdoor public sports events include running and cycling events listed on http://hlaup.is; http://hri.is.

likely reason for the absence of the mature plume from the public advisories was that the spatial and temporal (72 h) domains of the operational forecasting model were not sufficiently large to capture the events where the plume returned to Iceland[7,9,10]. The occasions where the mature plume was successfully identified in the forecast were likely coincidental: there was likely mixing in Reykjavík of a primitive plume transported directly from the eruption site, and a separate, mature plume being advected from outside of the modelling domain.

**Respiratory disease associated with mature plume**. We considered the impact of the mature plume exposure on the daily number of AMD (daily-mean number and standard deviation of individuals 131 ± 70), PCMD visits related to respiratory disease (142 ± 63), and HED visits related to respiratory disease (18 ± 8) using regression analysis (see Methods). The results— 'effect estimates'—are reported as percent change in relative risk (RR) compared to a scenario in which all other variables are held constant (Figs. 2a–c). In order to establish whether SO$_2$ gas is the causal agent for health effects of the mature plume we included SO$_2$ as a continuous variable covariate in the model in separate analyses (see "Methods"). We found that the health effects in the SO$_2$-adjusted results were typically lower than the unadjusted results by 0.5–10.0%, but still estimated a statistically significant ($p < 0.05$) increase in most health outcomes (Fig. 2d–f). This suggests that there may have been residual effects associated with

the mature plume which could not be attributed to SO$_2$ alone. Other potentially confounding factors were analysed in "Methods".

In the SO$_2$-unadjusted analysis, total AMD increased by 19.3% (95% confidence interval (CI) 9.6–29.1%) on the day of exposure (lag 0). When separated into age groups, the lag 0 increase was 28.0% (95% CI 13.6–42.4%) in children (<18 years old); by 18.2% (95% CI 6.9–29.6%) in adults (18–64 years old); and 15.6% (95% CI 3.4–27.8%) in elderly (>64 years old). At lags 0–1 and 0–2, the effect estimates were somewhat attenuated but still reached statistical significance (Fig. 2a). After adjusting for SO$_2$, only lag 0 was increased significantly in adults and elderly (Fig. 2d).

PCMD were increased at lag 0 in adults by 10.9% (95% CI 4.3–17.5%); and 27.9% (95% CI 13.3–42.6%) in elderly; no significant changes for children. At longer lags there were no significant associations (Fig. 2b). After adjusting for the presence of SO$_2$, the results were largely similar (Fig. 2e).

For HED in adults and elderly there were no significant associations with exposure at lags 0–1, but they were significant at lags 2–4. For HED in children, lags 0–2 was significantly associated with exposure (Fig. 2c). HED visits were increased at lag 0 and lags 0–1 in children by 27.4% (95% CI 7.5–47.2%). In adults, HED visits at lag 2–4 were increased by 27.7% (95% CI 12.7–42.7%). After adjusting for SO$_2$, HED visits in children remained increased at lag 0 and lags 0–1; in adults, effect estimates remained increased at lag 2–3 and lag 3–4 (Fig. 2f).

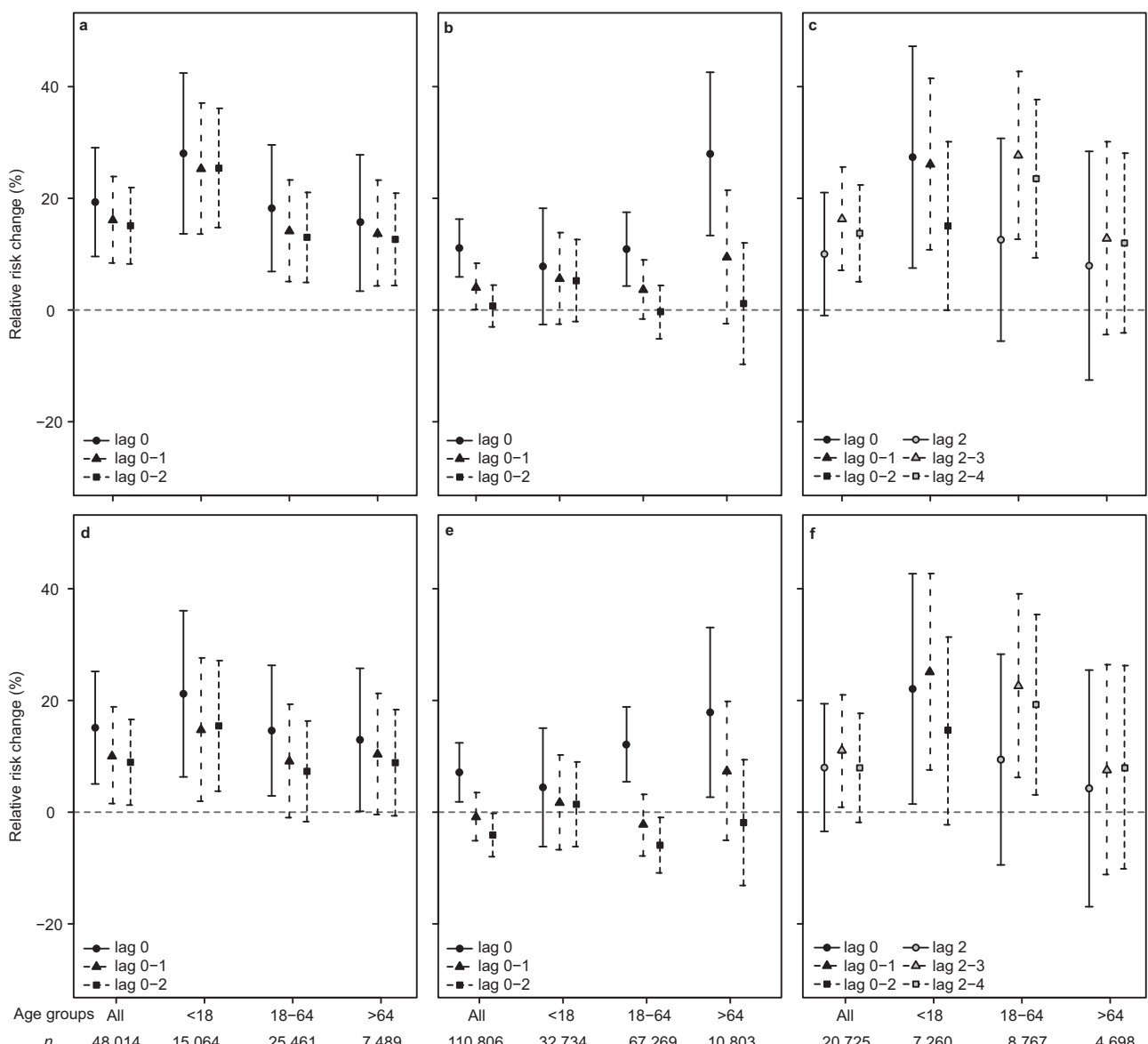

**Fig. 2 Mature plume exposure and respiratory health outcomes.** Associations between mature plume exposure and respiratory health outcomes from regression analysis for (**a**, **d**) AMD, $n = 48,014$ individuals; (**b**, **e**) PCMD, $n = 110,806$ individuals; (**c**, **f**) HED, $n = 20,725$ individuals. **a–c** Unadjusted results. **d–f** Results adjusted for $SO_2$. Results are reported as percent change in RR with error bars showing the 95% confidence intervals. Results are shown for different age groups (all age groups combined; <18 years; 18–64 years; >64 years) for several lag combinations; $n$ of individuals in each age group is shown on the x-axis. Lag 0 = the day of the exposure, lag 1 = one day after the exposure, lag 2 = two days after the exposure, lag 0–2 = mean of lag 0, 1, and 2, lag 2–3 = mean of lag 2 and 3, lag 2–4 = mean of lag 2, 3 and 4. On (**c**, **f**) the association between exposure and HED visits are shown for combinations of lag 2–4 in adults and elderly; in children, combinations of lag 0–2 are shown as there was no significant change for other age groups.

In agreement with our results, Longo et al.[22] reported elevated rates of respiratory symptoms in Hawaii high-vog zones for all age groups, based on analysis of a small number of total visits (mean < 10 per day) to local clinics and emergency rooms. While we have no direct comparison of HED during other large volcanogenic pollution events, a possible analogue is the anthropogenic London fog event of December 1952–1953, where hospital bed application for respiratory disease indicators peaked several days after the maximum $SO_2$ concentration[35], similar to our HED results for adults and elderly if it is (reasonably) assumed that a peak $SO_2$ in London fog was associated with a peak in $SO_4^{2-}$.

**Comparison of mature and primitive plumes.** The health effects associated with exposure to (i) primitive plume and (ii) mature plume were compared in models configured to adjust for both

types of exposure, Fig. 3 (see "Methods"). In a time series analysis, events of different types that occur close in time can be a confounding factor. We used a conservative method for coding exposure days (0 or 1) in the main generalized additive models (GAM) analysis, which would lead to an underestimate of the effects rather than an overestimate. Autocorrelation can cause results to be biased, but plots of the autocorrelation function of the model residuals were inspected and found to be acceptable (within the 95% confidence interval). The effect estimate of the primitive plume was predicted when $SO_2$ concentrations exceeded the air quality exposure threshold (24-h mean >125 µg/m³). We cannot exclude that lower levels of $SO_2$ were associated with respiratory health outcomes; this was not analysed here.

There were no significant differences in effect estimates between the adjusted primitive plume (Fig. 3) and previously

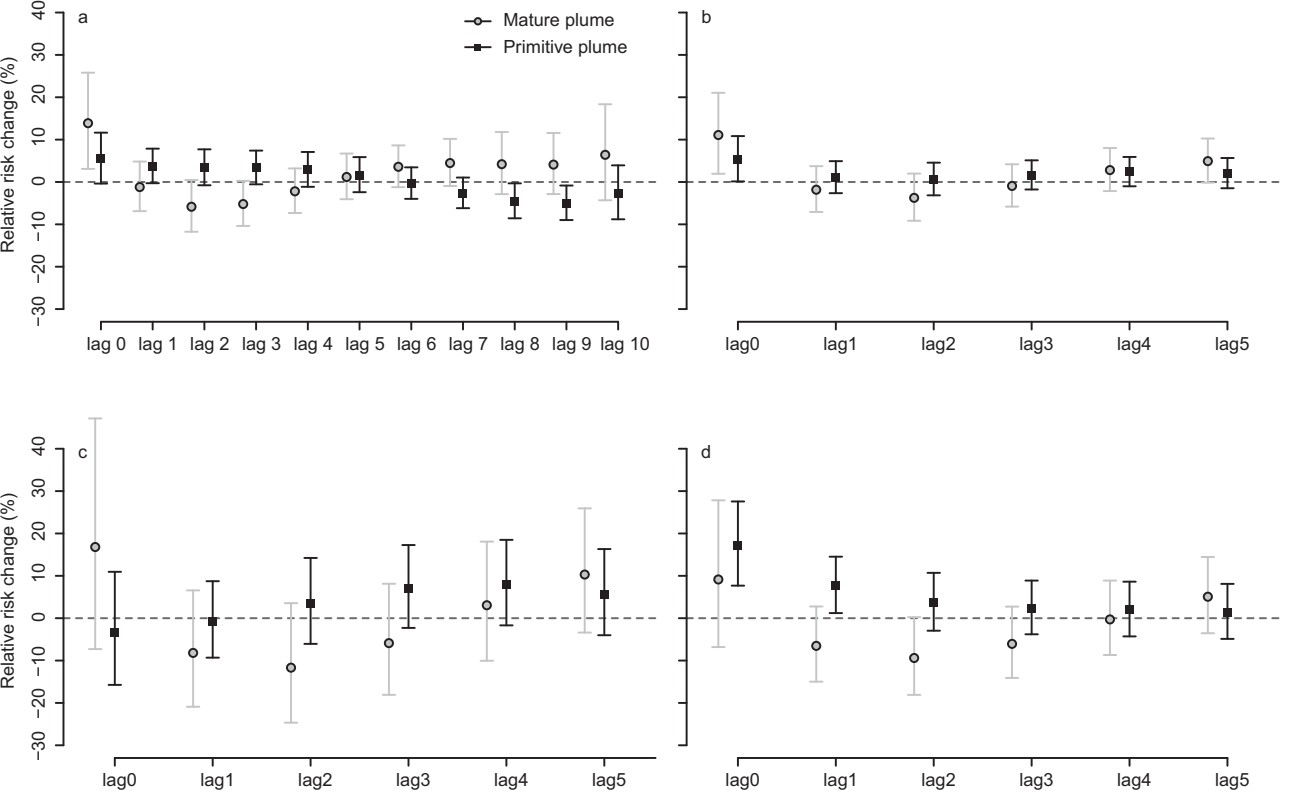

**Fig. 3 Comparison of mature and primitive plumes.** Associations at different time lag between exposure to mature and primitive plumes (where lag 0 is the day of the exposure, lag 1 is one day after the exposure, and so on) for (**a**) AMD (n = 48,014 individuals with dispensings), and (**b–d**) respiratory diagnosis categories in PCMD (ICD codes and n of individuals in brackets). **b** Infectious respiratory diseases (J00–J22, n = 89,406 individuals); **c** Chronic bronchitis and sinusitis (J30–J39, n = 8706 individuals); **d** Asthma/chronic obstructive pulmonary disease (COPD) (J44–45, n = 8065 individuals). Results reported as percent change in RR with error bars showing 95% confidence intervals from a two-pollutant distributed lag regression model.

published effect estimates of a primitive plume that did not take the mature plume into account[31]. This suggests that the two types of plume, primitive and mature, had independent effects on respiratory health outcomes.

AMD and PCMD for respiratory infections were significantly increased at lag 0 for both the primitive and mature plumes (Figs. 3a, b). In the absence of an influenza epidemic (see "Methods") this suggests a link with volcanogenic pollution. Association with the mature plume showed a higher increase than the primitive plume, although there is a large overlap in the confidence intervals. PCMD asthma and COPD were significantly increased at lag 0 only when associated with primitive plume exposure (Fig. 3d). Other respiratory subcategories—PCMD chronic bronchitis and sinusitis (Fig. 3c)—did not reach statistical significance, although some increase was seen associated with mature plume exposure at lag 0.

The observed differences in the effects associated with primitive and mature plumes may be related to the differences in their respective chemical compositions. Relating the observed morbidity for asthma and COPD (Fig. 3a, c) to the previous literature, experimental studies have found that exposing individual with asthma to both sulphuric acid ($H_2SO_4$) and $SO_2$ was associated with reductions in lung function[36]. In another experiment[37] where volunteers with and without asthma (results combined in the original study) were exposed to sulphuric acid, respiratory symptoms were increased. Increases in infectious respiratory diseases are concurrent with in vitro experiments of exposure to sulphuric acid reducing the immune system response of pulmonary cells[38].

It is also possible that the higher effect estimates associated with exposure to the mature plume may at least partially be

attributed to the more successful forecasting of the primitive plume (Table 1). We examine the effect of public advisories in the next section.

**Effect of public advisories on health care usage.** Every advection event of the primitive plume was forewarned in public advisories during the study period, in contrast to only half of the mature plume events (Table 1). We analysed the health effects separately for the no-advisory mature plume days (Fig. 4, Methods) to test whether public advisories had a measurable impact on the registered health care usage.

Generally, the AMD effect estimates were lower when no-advisory days were included in the analysis (by 9.5–16.2% RR, Fig. 4a). On the contrary, PCMD effect estimates were higher when no-advisory days were included, especially for the elderly age group (Fig. 4b). HED were higher at longer lags (lags 2, 3 and 2–4) when no-advisory days were included, with the elderly age group showing the largest difference (Fig. 4c). Effect estimate on HED in children did not reach statistical significance and is inconclusive (Fig. 4c).

These results indicate that public advisories changed health behaviours among the population. An increase in AMD associated with the issuing of public advisories suggests that patients were taking precautionary measures by stocking up on medication. On the other hand, where mature plume days were not included in a public advisory, there are increased outcomes for HED visits, and to some degree PCMD, particularly in the elderly age group. This suggests that public advisories were useful for protecting vulnerable individuals, and that HED and PCMD may be truer proxies for acute health care needs than AMD.

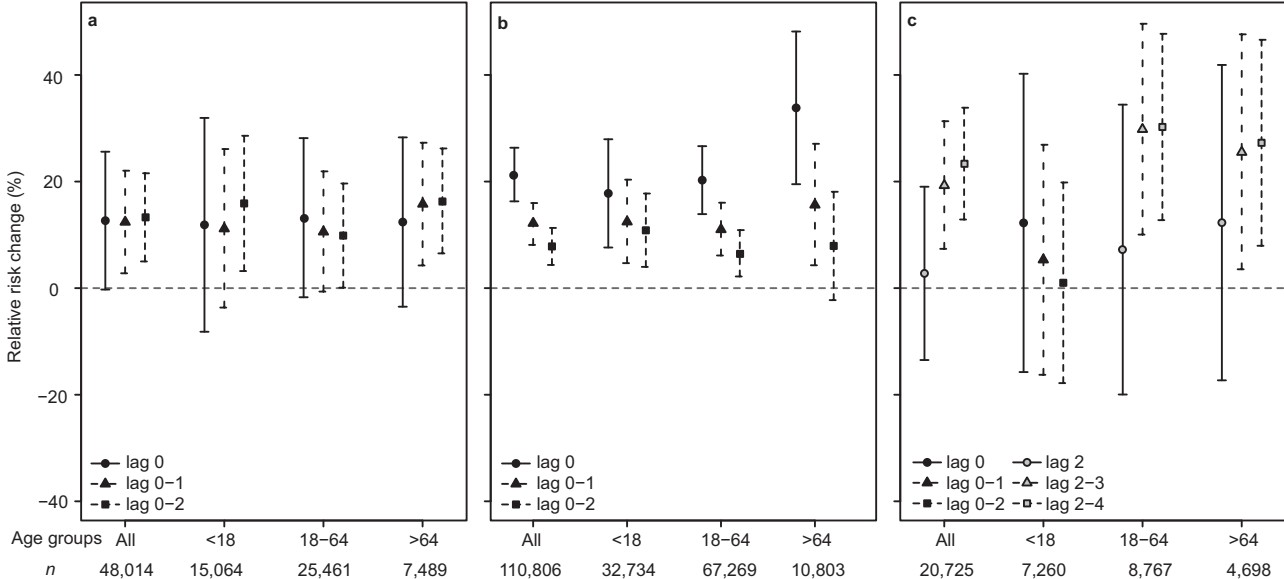

**Fig. 4 Mature plume exposure and respiratory health outcomes on no-advisory days.** Associations between mature plume exposure and respiratory health outcomes from regression models on days when no public advisories were issued for (**a**) AMD, $n = 48,014$ individuals; **b** PCMD, $n = 110,806$ individuals; **c** HED, $n = 20,725$ individuals. Results are reported as percent change in RR with error bars showing the 95% confidence intervals. Results are shown for different age groups (all age groups combined; <18 years; 18–64 years; >64 years) for several lag combinations; $n$ of individuals in each age group is shown on the x-axis. Lag 0 = the day of the exposure, Lag 1 = one day after the exposure, Lag 0 = the day of the exposure, lag 1 = one day after the exposure, lag 2 = two days after the exposure, lag 0–2 = mean of lag 0, 1, and 2, lag 2–3 = mean of lag 2 and 3, lag 2–4 = mean of lag 2, 3 and 4. On (**c**), the association between exposure and HED visits are shown for combinations of lag 2–4 in adults and elderly; in children, combinations of lag 0–2 are shown as there was no significant change for other age groups.

**Non-respiratory health impacts**. We analysed the association between exposure to the mature plume and PCMD and HED for nausea and vomiting (ICD code R11), headaches (R51) diseases of the circulatory system (ICD codes starting with I), and eye diseases (H10–H11), see "Methods". The results were non-significant in most cases and are included in Supplementary Figs. 1 and 2.

Statistically significant increase was seen in PCMD circulatory system disease in children at lag 0 after adjusting for primitive plume (Supplementary Fig. 1). In HED, nausea and vomiting were increased in children and adolescents at lag 0–1 and lag 0–2 (Supplementary Fig. 2). There was little consistency between PCMD and HED outcomes and the number of daily cases were very low (except for circulatory system disease), limiting the statistical power as is evidenced by the wide confidence intervals. We refrain from making further analysis of these outcomes in the health register data.

Nausea and headaches were present as self-reported symptoms by members of the public in Iceland. An online questionnaire was made available by the IMO during the eruption and comprised of 4 yes/no questions about the presence of the smell of sulphur, irritation of the eyes, throat, nausea, as well as a free-form comments box for other symptoms. A total of 138 responses were received during the eruption, with a single-day maximum of 30. Days impacted by a primitive plume received a greater number of responses than mature plume days (Fig. 5). For the mature plume days, the number of responses was higher on no-advisory days. Nausea was reported in a total of 34 responses (25%), and 3 responses (2%) noted headaches (reported under 'Other symptoms'). It is important to note that self-reporting by self-selected participants is prone to bias. However, the results may suggest that the health impacts of volcanogenic air pollution may have been more varied than identified in the register-based analysis. This is potentially due to the fact that people are relatively unlikely to present to paid healthcare services when experiencing mild or transient symptoms.

**Causal mechanisms**. As previously discussed, the arrival of the plume in Reykjavík did not cause a large increase in PM concentrations compared to the pre-eruption background. The mean concentrations ($PM_{2.5} < 9 \, \mu g/m^3$) remained within good air quality guidelines, and the peak concentrations ($PM_{2.5}$ 18 $\mu g/m^3$) were much lower than those observed during the pre-eruption study period ($PM_{2.5}$ peak 160 $\mu g/m^3$)[10]. However, the associated increases in AMD, PCMD and HED were significant in our analysis, and they were also large when compared to those associated with increased PM concentration from other sources in Iceland[39]. The other sources included explosive volcanic eruptions (Eyjafjallajökull 2010 and Grímsvötn 2011), dust storms and traffic pollution[39]. This suggests that exposure to increased PM concentration cannot alone explain the observed health impacts. The increase cannot be fully attributed to $SO_2$ exposure either, as previously discussed (Fig. 2). The chemistry of the PM in the mature plume may therefore be at least one of the causal factors. Here we review the available evidence to hypothesise about potential mechanisms by which the mature plume may cause the observed health impacts.

A mature volcanic plume is a highly heterogeneous mixture of sulphate (typically the dominant component by mass) and a large range of trace components, including metal pollutants (e.g. lead, arsenic and cadmium)[10]. Sulphate can be found as sulphuric acid ($H_2SO_4$) or as a sulphate salt. Toxicological studies on exposure to neutralised and relative benign forms of sulphate (e.g. ammonium sulphate, $NH_4SO_4$) predict no measurable health impacts (reviewed by WHO[40]). However, controlled exposure studies of sulphuric acid in fine mode PM show a tendency for adolescent asthmatics to be more sensitive than adults and elderly[36], which is in line with the higher effect on asthma medication for children observed in our study. In studies of controlled exposure of humans to coarse mode sulphuric acid ($\geq 2.5 \, \mu m$) at significantly higher levels than observed in our study (>500 $\mu g/m^3$ for one hour), an effect was seen on increased

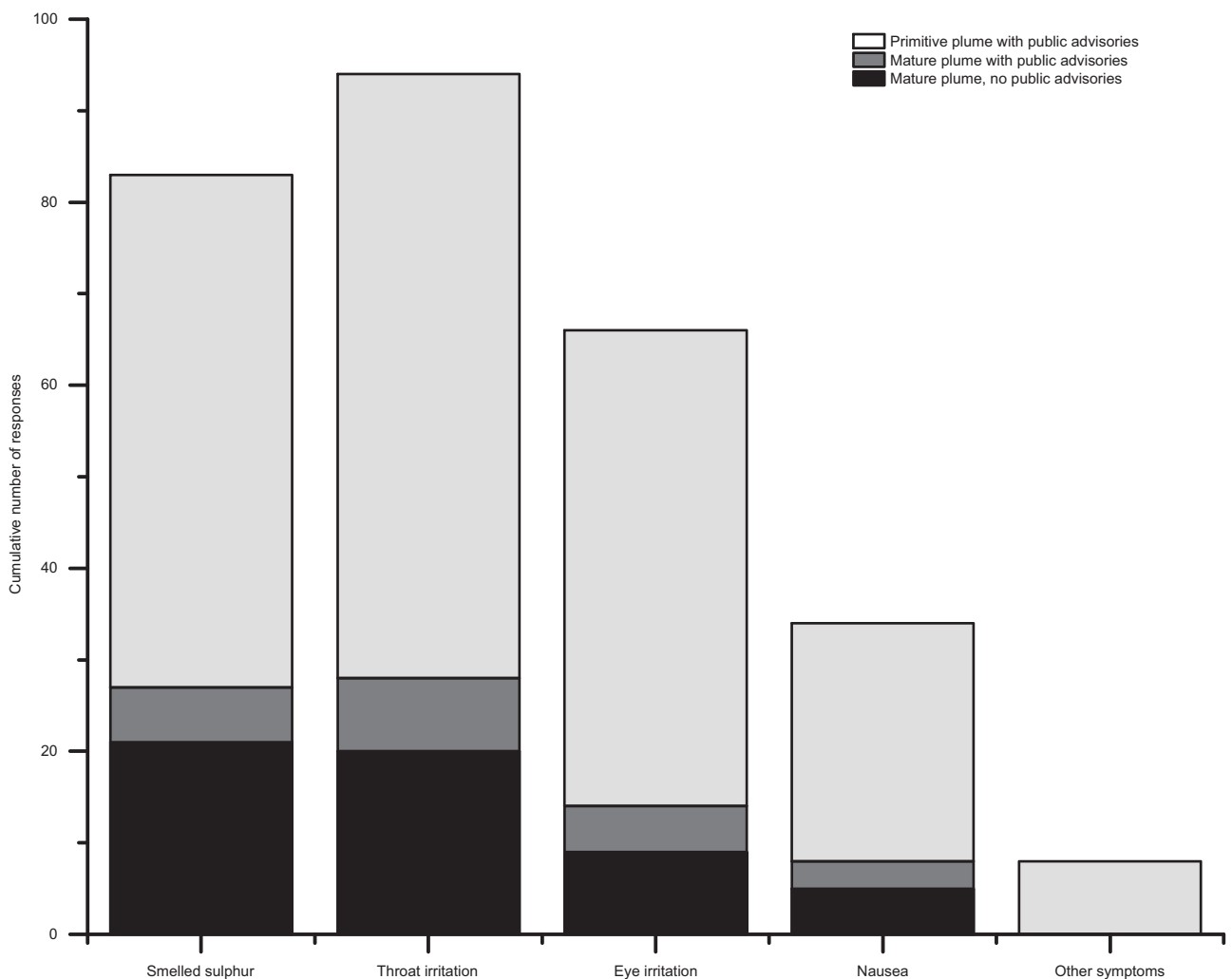

**Fig. 5 Self-reported symptoms in Reykjavík.** The figure shows the cumulative number of responses to the IMO online questionnaire. The responses are separated into those recorded on days impacted by a primitive plume (always associated with a public advisory), by a mature plume associated with public advisories, and by a mature plume not associated with a public advisory. 'Other symptoms' were recorded in free-form text box and included, e.g., headaches and increased asthma symptoms.

respiratory symptoms[37]. In animal studies (rabbits), controlled exposure to concentrations of fine mode (≤2.5 μm, most between 0.1 and 1 μm) sulphuric acid by at least 50 μg/m$^3$ for 4 h or 100 μg/m$^3$ for two hours also altered the pulmonary immune system clearing mechanisms[41], an effect which worsened with daily exposure over several months[42]. Immune system responses, TNF-α and reactive oxygen species, were significantly depressed 24 h after exposure to 750 μg/m$^3$ sulphuric acid[38].

It has also been proposed that sulphate may be only a marker for causal agents, or that toxic species may be formed in its chemical reactions with trace metals[43,44]. As reviewed by Reiss et al.[45], sulphate components in an air pollutant mix may contribute to health impacts indirectly by increasing the availability of metals. Sulphate has been reported to increase the solubility of iron in particles[43], while metals may catalyse the production of sulphuric acid which in turn might react to produce toxicologically active soluble metal salts. The exact toxic mechanisms by which metals act are unknown, but free radical generation has been proposed by many (e.g.[46]). Lippmann[47] has also argued that sulphate represents the end stage of a chain of reactions that produces toxicologically active species, and that the hydrogen ion (acidity) is a key factor. Lippman[47] also stresses the potential importance of copper, nickel and vanadium, but

the causal evidence for these and for other metals remains limited.

Epidemiological studies on $SO_4^{2-}$ exposure are relatively scarce and largely focus on anthropogenic pollution (e.g.[33,45,48]). Exposure to sulphuric acid PM has mainly been associated with short term effects on cardiovascular outcomes[47], but few studies have found increases in respiratory outcomes. This contrasts with our results, and the results of vog exposure studies in Hawaii. In epidemiological studies of vog exposure, health effects included self-reported cough and possibly an increased risk of impaired lung function (low $FEV_1/FVC$ ratio) in 9–11 year olds[24,25]. Following an increase in volcanic emissions and associated air pollution in Hawaii in 2008, the clinics and emergency rooms of the surrounding areas experienced increased visits due to respiratory causes for all age groups[22]. Survey and interview studies of adults have identified increased rates of self-reported bronchitis, sinus pain, and several other symptoms[23,26].

There is a large difference in the formation processes and the resulting composition of PM in anthropogenic pollution from fossil fuel combustion (the subject of most of the health effect literature), and in volcanic plumes where PM is largely formed through condensation of magmatic gases. PM formation mechanisms and composition are key controls on the solubility

of the chemical constituents[11,14,49], which in turn may be an important, but a poorly understood factor in the observed health effects[50]. This limits the overall applicability of risk estimates from anthropogenic to volcanic pollution studies, and emphasizes the need for further investigations specific to volcanic environments.

**Societal context and recommendations**. The impact of the Holuhraun eruption on air quality and public health should be discussed within the societal context of Iceland, a high-income country with good access to healthcare that is subsidised by the government[51]. Due to the high-latitude climate (64 °N in Reykjavík), homes are predominantly well-built concrete structures with double-glazed windows, providing good protection from outdoor air pollution, although windows are typically kept continuously open for ventilation, encouraged by the cheap geothermal heating[52]. While Holuhraun public advisories recommended reducing the amount of time spent outdoors and closing windows during the volcanogenic pollution events, we showed that approximately half of the mature plume pollution events were not forecast (Table 1).

Holuhraun eruption occurred during autumn and wintertime when limited daylight hours and more frequent occurrences of adverse weather act to reduce the number of time people spend outdoors, and therefore the amount of potential exposure. However, the number of people who spent prolonged time outdoors was likely non-trivial. In particular, in Iceland it is commonplace to let babies and infants nap in a pram outdoors, and school-age children typically spend the recess outside. This is a concern as our results show that children were a vulnerable group. Iceland is also a hugely popular destination for nature-oriented tourism and received a total of ~270,000 visitors during the eruption in September–December 2014[53], almost doubling the population of Iceland (~330,000[54]) and thereby the number of potentially exposed people. Furthermore, there were at least 30 outdoor public sports events across Iceland (each attracting up to 500 participants) during the eruption study period; seven of these events in Reykjavík coincided with the days impacted by the mature plume (Table 1). While our data cannot be used to conclude whether exercise exacerbates the negative health effects of a mature plume, and no studies have looked specifically on exercise-related effects of volcanic air pollution, several studies have found that asthmatics responded to sulphuric acid exposure only during exercise[36]. Aerobic exercise increases the ventilation rate-making athletes more exposed, and some air pollution types have been found to be detrimental to lung function, especially in individuals with underlying respiratory diseases[55].

Although volcanic emissions cannot be directly controlled, their impact can be reduced through hazard characterisation, and risk assessment and mitigation. In addition to the large but relatively short-lived fissure eruptions, there are tens of volcanoes worldwide which are emitting gases and PM at a lower rate but for much longer periods (e.g. Masaya in Nicaragua 1993-ongoing). We recommend that for volcanic events which impact air quality at ground level, public advisories for both primitive and mature plumes are issued to protect the vulnerable groups. Our observational study cannot demonstrate whether sulphate was the causal agent for the observed health impacts, or simply a marker for one or more toxic constituents in the mature volcanic plume. However, sulphate formed through $SO_2$ gas-to-PM conversion is relatively simple to include in plume dispersion simulations and could be used as a proxy for mature plume advection into populated areas. The health impacts observed by us were associated with exposure to a plume with relatively low sulphate concentrations (3–8 μg/m³). In the absence of evidence

for a safe concentration of sulphate and/or other PM components in the mature plume, the public advisories should be based on a simple 'plume present' message using a single threshold. The threshold of 5 μg/m³ of sulphate suggested previously by Lioy and Waldman[33] is too high for this purpose, and a lower value (<1 μg/m³ above the non-volcanic background) would be more appropriate. Our recommendations are in line with those of 'Review of evidence on health aspects of air pollution project' by World Health Organisation[40], and would enable the public to take necessary steps to avoid exposure or mitigate the risk.

Further studies are also recommended to investigate if the exposure to the volcanogenic pollution from Holuhraun is associated with longer-term health effects in addition to the acute effects reported here.

## Methods

The study period was 1 January 2010 to 31 December 2014, which was selected based on the availability of health records. The Holuhraun eruption took place between 31 August 2014 and 27 February 2015. The time before the eruption was used a reference period. The study period covers only 4 out of the 6 months of the eruption due to a change in the database recording of events. This is not considered to significantly impact the conclusions of this study. The eruption intensity waned significantly after the first 4 months[20] and only 5 out of the 38 volcanogenic pollution days occurred in January–February 2015[10].

The mean population of Iceland during the study period was ~330,000 inhabitants. The case study area is Greater Reykjavík, Iceland's capital region which comprises the capital city Reykjavík and 6 municipalities around it. In this study, it is collectively referred to as Reykjavík. Reykjavík had 205,282 residents at the beginning of the study period, and 215,965 residents at the end[54]. This was the first time a population of this size and density (500 per km²) was assessed for health impacts of volcanic PM pollution using a population-wide register.

**Air quality data**. Time-series measurements of $SO_2$ and $SO_4^{2-}$ were collected by the EAI air quality stations. Data were provided to the researchers by EAI as Microsoft Office Excel files and all subsequent data analysis was performed in Microsoft Office Excel 2010. The data analysed in this work were collected at two Reykjavík stations, Hvaleyrarholt (HVAL) and Grensásvegur (GREN). HVAL monitors concentrations of gaseous pollutants including $SO_2$ (up to 1 min time series resolution), $PM_{2.5}$, and sulphate PM (by filter sampling, 24-h mean value). HVAL was used to calculate the ratio of $SO_4^{2-}$ to $SO_2$ in the plume to identify whether it was primitive or mature. GREN monitors only the gaseous pollutants and $PM_{2.5}$, but it was considered to be more representative than HVAL of the population exposure to $SO_2$ due to its location (central vs rural, respectively). GREN data were therefore used for calculating the health effects associated with $SO_2$ exposure. There was a good correlation between $SO_2$ time series at HVAL and GREN: Pearson's correlation coefficient = 0.966, $p < 0.001$ during the eruption. The $SO_2$ concentrations at GREN were typically higher than at HVAL, with 10 exceedances of 24 h-mean $SO_2$ air quality guidelines versus 8, respectively, during the eruption study period (Table 1).

The period around New Year's Eve (29 December to 1 January each year) was excluded from all analysis due to the noticeable impact on the concentration of the analysed air pollutants sourced from fireworks and bonfires. Further details on instrumentation and analytical techniques can be found in ref. [10].

**Health care usage**. The Icelandic health care system is state-centred, mainly a publicly funded system with universal coverage[51]. All relevant ethical regulations were complied with for use of health records in this study. We obtained data on respiratory health and individual data on residence (postcode), age, sex and an anonymous personal identification number from (1) the National Medicines Register; (2) Primary care centres (that function as first point of contact) and (3) Landspítali, the national university hospital, the country's centre of clinical excellence[51]. All registers are held by the Icelandic Directorate of Health. The Icelandic Bioethical Committee approved the extraction for the use in this study (ref. no. for the current study: VSNb2015050022/03.01). In compliance with the relevant ethical regulations, informed consent was not obtained from individuals.

From the National Medicines Register we extracted data on dispensing (pharmacy sales to individuals) of prescription anti-asthma medication (AMD) classified by The World Health Organisation Anatomical Therapeutic Chemical code R03. AMD relieve symptoms of asthma and chronic obstructive pulmonary disease, COPD, and are occasionally prescribed to individuals with respiratory infections. Furthermore, AMD is a proxy for respiratory health in a population[56–58]. From the primary care centres (PCMD) and hospital emergency department (HED) databases at the Directorate of Health we extracted data on individuals diagnosed with respiratory diseases.

In our analysis we consider the number of in-person visits to PCMD and HED (regardless of HED admission status). As the same bout of illness is likely to result

in recurring contacts with the health care system, we included only the first record of an individual's visit within a 14-day period for the same diagnosis category to avoid exposure misclassification with respect to the timing of the outcome. For each outcome, we constructed daily time series starting 1 January 2010 to 31 December 2014 for the following age groups; children (under 18 years of age), adults (18–64 years), and elderly (age 65 years and above). For PCMD visits, we present results for several categories of respiratory disease (1) infectious diseases including acute upper respiratory infections (ICD[59] codes J00–J06), influenza and pneumonia (J09–J18), and other acute lower respiratory infections (J20–J22), (2) Rhinitis, sinusitis, and other diseases of the upper respiratory tract (J30–J39), and (3) obstructive respiratory disease, including chronic obstructive pulmonary disease (COPD) and asthma (J44 and 45). For non-respiratory diagnosis categories, we analyse PCMD and HED visits for circulatory system disease (ICD code I), nausea and vomiting (code R11), headaches (code R51), and conjunctivitis (code H10-H11). Age-stratified results (<18, 18–64, >64 years) are also analysed and presented.

The mean number of daily AMDs during the study period (1 January 2010–31 December 2014) was 131 (SD 70). The mean number of daily PCMD visits which were assigned respiratory health diagnoses was 142 (SD 91), and mean number of daily HED visits which were assigned respiratory health diagnoses was 18 (SD 8). The mean number of daily PCMD visits for non-respiratory diagnoses was: nausea and vomiting 1.1 (SD 1.1); headaches 1.9 (SD 1.5); circulatory system disease 21.2 (SD 10.9); eye irritation 1.2 (SD 1.1). The mean number of daily HED visits for non-respiratory diagnoses was: nausea and vomiting 2.7 (SD 2.4); headaches 9.4 (SD 6.6); circulatory system disease 130 (SD 90); eye irritation 18 (SD 7.7).

**Statistical methods.** To estimate the effect of the mature plume, the number of daily events were regressed on an indicator value for the exposure value in time series regression models[60]. The exposure indicator variable was coded "1" for days with mature plume and "0" for days without mature plume. In the distributed lag analysis where more days were considered at once, the presence of one exposed day during a two- or three-day period qualified the period as exposed. All statistical analysis was performed in R studio desktop (version 1.2.5), R (version 3.6)[61] using the "mgcv" library (version 1-8.27)[62].

*Determining the delay from exposure to health outcome.* There can be a time lag from the exposure (a day with high pollutant levels) to the observed health outcome (individuals seeking health care or buying medication). We investigated the presence of time lags using splines and distributed lag non-linear model (DLNM)[63] method, Supplementary Fig. 3. For AMD and PCMD we report the association between exposure on the same day as the outcome (lag 0); cumulative exposure on the day of the exposure and the day after (lag 0, 1); and cumulative exposure on the day of the exposure and the two following days (lag 0–2). For HED visits in adults and elderly there were no significant associations at lags 0, 1 but there were significant associations at lags 2–4. In HED visits in children, only lags 0–2 were significantly associated with the exposure.

*Estimating the effect of exposure to mature plume using GAM models at short (0–4) lags.* The models used here were GAM with Poisson distribution because the outcome is in counts. Quasi-Poisson distribution, a more appropriate distribution for data with some heteroscedasticity, was tested and used when the residual diagnostics indicated it to be a better fit (which was the case for AMD). We estimated the effects of SO₂ exposure on the outcome by fitting GAM[64] as shown in Eq. (1):

$$Y_t \sim \text{Quasipoisson}(\mu_t)$$
$$\log \mu_t = \alpha + \beta_1 \text{Mature plume indicator} + \beta_2 I_{\text{dow}} + \beta_3 \text{Strike} + \beta_8 Y_{t-1} \quad (1)$$
$$+ s_4(\text{Day in the time series}) + s_5(\text{Day of the year, bs} = ''\text{cc}'')$$

where $Y_t$ denotes the daily number of health events, $\alpha$ is the intercept, $\beta_1$ denotes the log relative rate of events associated with the presence of the mature plume. $I_{\text{dow}}$ is an indicator for a day of week and odd holidays (as the distribution of counts of daily AMD, PCMD and HED are bimodal as there are fewer visits during weekends). Strike is an indicator of strike days (used only in analysis of the HED and PCMD data to indicate days where medical doctors went on strike as part of a labour conflict, which coincided with some exposure days). We used an autoregressive term (adjusting for the outcome at lag 1, $Y_{t-1}$), to improve the autocorrelation of the model residuals. The models were adjusted for time trend to account for an increasing population during the years 2010–2014, and seasonality using splines[31]. The results—estimates of relative risk (RR)—were reported as percent change in RR compared to a scenario in which all other variables are held constant. 95% confidence limits are reported.

Furthermore, we constructed GAM models that were adjusted for continuous 24-h daily SO₂ concentrations to investigate if the associations between mature plumes and the health outcomes were confounded by the presence of SO₂.

To investigate the effect of public advisories, we followed the same methods as described above but considered only the days with no public advisories.

*Estimating effects of combined exposure and delayed effects with DLNM models.* Because the days impacted by the mature plume were often temporally close to

primitive plume days we analysed the two exposure types in the same model and estimated the effects and time lags up to 10 days for AMD, and six days for PCMD outcomes with DLNM[63]. High SO₂ concentrations (24-h mean ≥125 µg/m³) were used as a marker for primitive plume presence. See details in Supplementary Methods. This analysis was run for AMD and diagnostic subcategories of respiratory health in PCMD (Fig. 3).

**Potential confounding factors of measured health effects.** We considered whether the flu season, MD strike, day-of-week effects and time trends could be confounding factors in the measured health effects. We found no impact of these factors on the effect estimate for the mature plume. The results were not adjusted for air pollutants PM₁₀ and NO₂ as the mature plume was highly heterogeneous in composition[10] and the correlation between these pollutants varied between the days with mature plume exposure. In multi-pollutant models, the estimated effects of the mature plume effects tended to be higher (by few percentage points, data not shown). There is evidence from in vitro toxicological studies that combined exposures to volcanic ash and urban pollutants (e.g. diesel exhaust particles) may have an increased impact on respiratory effects when compared to their individual exposures (likely an additive effect from the increase in PM concentrations)[65]. It is unclear whether the same would apply to ash-poor volcanic plumes, as volcanic ash (formed by magma fragmenting) is typically significantly coarser and less water-soluble than sulphate and other particles formed through gas-to-PM conversion (e.g.[11,13,14]). Therefore we refrained from analysing these results further in this study. We do not consider this to impact our results because, as mentioned in the main text, the observed increases are higher than those associated with other PM pollutant types[39] and there were no above-background PM₂.₅ pollution events during the eruption study period, showing that there were no significant PM₂.₅ events from non-volcanic sources.

**Reporting summary.** Further information on research design is available in the Nature Research Reporting Summary linked to this article.

## Data availability

The health record data that support the findings of this study can be made available to researchers after approval of a formal application to the Icelandic Directorate of Health and the Icelandic Bioethics committee. The data are not publicly available due to them containing sensitive individual-level information.The SO₂, PM₂.₅ and SO₄²⁻ data that support the findings of this study are available from the Environment Agency of Iceland but restrictions apply to the availability of these data, which were used under license for the current study, and so are not publicly available. Data are however available from the authors upon reasonable request and with permission of the Environment Agency of Iceland. The results of the Icelandic Meteorological Office online questionnaire that support the findings of this study are available from the Icelandic Meteorological Office but restrictions apply to the availability of these data, which were used under license for the current study, and so are not publicly available. Data are however available from the authors upon reasonable request and with permission of the Icelandic Meteorological Office.

## Code availability

RStudio desktop (version 1.2.5), R (version 3.6)[56], library "mgcv" (version 1.8-27)[62] was used for statistical analysis of health care records. Example of the R Studio codes created by the authors are provided in the Supplementary Methods. All R Studio codes created by the authors are available upon reasonable request.

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

## Acknowledgements

Prof. RL Maynard contributed to the discussion. E.I. acknowledges NERC urgency grant NE/M021130/1 'Source and longevity of sulphur in Icelandic flood basalt eruption plumes', GCRF UNRESP (NE/R009465/1, NE/P015271/1) and NERC COMET. H.K.C. acknowledges a grant from the Icelandic Research fund (RANNÍS, 152587051). A.S. acknowledges NERC V-PLUS (NE/S00436X/1). Figure 1 and Supplementary Movie 1 were originally published in Ilyinskaya et al.[10], Earth and Planetary Science Letters, https://doi.org/10.1016/j.epsl.2017.05.025 under Creative Commons License https://creativecommons.org/licenses/by/4.0/. Figure 1 was modified from the original version to show the location of the study area in this research. Supplementary Movie 1 has not been modified from the original version.

## Author contributions

The first two authors, H.K.C. and E.I. contributed equally to the development of this study and led the writing of the manuscript. H.K.C. analysed public health data and produced Figs. 2–4. E.I. analysed the air quality data, public observations, and produced Fig. 5. T. Gudnason and H.B. provided health care data. R.G.F. and T.J. provided EAI air quality data. R.G.F., F.D., U.A.V., T.A. and T.T. participated in the public health data analysis. P.J.B., H.B., T. Gislason, A.S., S.B. and M.A.P. contributed to the discussion and interpretation of the results. All authors commented on the manuscript drafts.

## Funding

## Competing interests

The authors declare no competing interests.
