## [Peer Review File · Nature Communications]

REVIEWER COMMENTS

Reviewer #1 (Remarks to the Author):

This is a well written paper which provides useful data for the assessment of the health effects of exposure to volcanic ash (represented by use of health care facilities and medication dispensing). Health effects associated with volcanic eruptions are an area of work where there are relatively few recent studies and this is a welcome addition to that body of work. The study comprises a substantial exposed population and is able to distinguish between days where there is exposure to mature ash plumes and where there is exposure to primitive ash plumes. The conclusions of the paper appear to be well justified based on the analysis and the data presented.

The paper would benefit from some restructuring to present a more cohesive thread through the results section. In particular the section on 'other' health endpoints reported in lines 151-161 seems out of place coming between Figure 2 and the text description of the results presented in that Figure from line 162 onwards. It is also possible that the text reporting of all the results shown graphically in Figures 2 and 4 are not necessary as the relevant patterns and values can be deduced with relative accuracy from the Figures. If they are to be retained I would suggest that presentation in a Table may be more to understand by the reader.

Some more detail on the meaning of the outputs of the statistical analysis would also be beneficial. Expansion of what exactly is the meaning of the 'relative risk' (relative to what...) and the 'percentage relative risk change' would be advantageous before presentation of Figure 2 and its associated data in the paper. It might be easier to interpret the overall results if Figures 2 and 4 were combined (as has been done in the explanatory text) so that there are 6 sub-figures within one overall figure.

Again the section lines 203-218 cross-refer to Figures 2 and 4 and would be better placed nearer that section with all the sections of self-reported health grouped together subsequently. As the authors note, there were relatively few responses to the self-reported symptoms questionnaire and these responses are likely to be prone to bias.

The lower effect seen on non-advisory mature plume days compared with advisory mature plume days is important as it implies that the fact of an advisory being issued may in itself change health behaviours among the population, as discussed in line 248.

Minor comments:

line 145 - can the units be specified i.e. assume the daily mean is the number of asthma medications dispensed, is this the number of people receiving medication or the number of items dispensed i.e. potential for a single individual to receive more than one item

throughout the paper there are a number of typos e.g.

line 39: should be 'were' not 'was'

line 46: some words missing before 'qualitative'?

line 114: 'dispensing' not 'dispending'

The paper needs careful proofreading.

Reviewer #2 (Remarks to the Author):

The findings of the study indicate an association between the exposure to mature (sulphate rich) volcanic plume and acute health effects in the exposed population, which has implications for the volcanic monitoring strategies, health risk assessment and population safety management in volcanic crises, globally. I trust the study is relevant for and should be of interest to, a wider scientific community (volcanologists, public health) and public (organisations) alike.

The study resembles some of the latest publications by the authors (e.g., Carlsen et al. 2019, under review at medRxiv) where the same approach has been used to correlate the air quality data with health care utilisation data for the same eruption (and study period). These data are unique in their high temporal resolution and their value has been well recognized, and acknowledged, in this paper; the study is justified and builds on the earlier work by focusing on specific events (mature plume days) that deserved more attention and in-depth analysis. Although the conclusions on the specific effects of mature plume remain somewhat reserved, the results demonstrate the differences in the recorded effects in days of the primitive and mature plume and thus show the need for consideration of their distinctive features and effects in the future studies. I consider the presented information important for advancing our understanding of the complexity of volcanic emissions and associated health effects, and specifically to serve for informing protection in volcanic areas that are not (well) monitored and lack robust data. Considering this, and that I see no major issues with the representation or discussion of the data, I kindly recommend this paper for publication.

The paper is overall well written, with a good balance of information and length. The statistical analyses seem sound and the methodology well-reasoned and sufficiently detailed, though, admittedly, I am no expert in such analyses. I added some (minor) comments/suggestions directly in the manuscript file, as I felt that additional clarification is necessary for certain parts of the text. Thanks very much to the Authors and Editors for your understanding and patience to receive this review in these uneasy times.

Very best wishes,
Dr. Ines Tomašek

Reviewer #3 (Remarks to the Author):

The authors examine the hypothesis that volcanic sulfur dioxide gas and sulfate aerosols are associated with increases in dispensing of anti-asthma medications, primary care visits and emergency department visits for respiratory complaints. The authors focus in particular on the effects of mature plumes of volcanic air pollution, as these sulfate-rich plumes are less well studied and, because they can be low in the criterion gas SO₂, may not be detected as a plume of volcanic air pollution to trigger an advisory.

The authors conclude that these sulfate-rich plumes are associated with increased anti-asthma medication dispensing, visits to primary care clinics and emergency departments for respiratory complaints.

The findings are not novel and are incremental to another preprint by the authors (<https://www.medrxiv.org/content/10.1101/19013474v1.full.pdf>) that more satisfactorily presents the methodology and details of the demographics and health record inquiry. Compared to the more fundamental finding of that preprint (that there was excess utilization for respiratory complaints during the eruption, compared to pre-eruption) this manuscript adds smaller insights about the relative lags in anti-asthma medications, clinic and ED visits. It could add much more to our understanding by discerning which symptoms or conditions are exacerbated by sulfate-rich aerosols, which may penetrate deeper into lower airways than hygroscopic SO₂ (wheeze and sputum production vs nasal congestion and eye irritation), or if there are susceptible persons, such as those with chronic obstructive lung disease, heart failure, smoking history.

Thus, the authors can provide more detailed methodology (or reference <https://www.medrxiv.org/content/10.1101/19013474v1.full.pdf>) and address additional

hypotheses given the availability of the register. (Compared to the health registry data, the self report questionnaire adds little substantial information.)

There are typographical errors, such as:

Line 39: was-> were

Line 56 low levels Less than 200 ppb?

Line 57 airways "rei"?

and more.

In Methods, this site is no longer

active?<http://vedur.maps.arcgis.com/apps/OnePane/azuretime/index.html?appid=da54a233cb0d4b679bdf5a1a860d11b9>

Shows no responses

Response to reviewers

Original text in black. Our response in blue.

Reviewers' comments

1. Reviewer #1

This is a well written paper which provides useful data for the assessment of the health effects of exposure to volcanic ash (represented by use of health care facilities and medication dispensing). Health effects associated with volcanic eruptions are an area of work where there are relatively few recent studies and this is a welcome addition to that body of work. The study comprises a substantial exposed population and is able to distinguish between days where there is exposure to mature ash plumes and where there is exposure to primitive ash plumes. The conclusions of the paper appear to be well justified based on the analysis and the data presented.

1.1. We thank the reviewer for their detailed and positive review of our work

The paper would benefit from some restructuring to present a more cohesive thread through the results section. In particular the section on 'other' health endpoints reported in lines 151-161 seems out of place coming between Figure 2 and the text description of the results presented in that Figure from line 162 onwards.

1.2. We have revised the results section in light of this to improve readability. We agree that the position of the figures relative to the text was not optimal, and apologise for the inconvenience caused. This was a page formatting issue. The resubmitted manuscript has the text and the figures+tables in two separate files.

It is also possible that the text reporting of all the results shown graphically in Figures 2 and 4 are not necessary as the relevant patterns and values can be deduced with relative accuracy from the Figures. if they are to be retained I would suggest that presentation in a Table may be more to understand by the reader.

1.3. We have revised the text describing the previous version of Figures 2 and 4 to make it more concise. We still believe that some text accompanying the figures is beneficial in order to highlight the most important findings so we decided to keep short text + figures, rather than figures + a table. Note that Figures 2 and 4 have now been combined into one based on the reviewer's next comment, and are now the new Figure 2.

Some more detail on the meaning of the outputs of the statistical analysis would also be beneficial. Expansion of what exactly is the meaning of the 'relative risk' (relative to what...) and the 'percentage relative risk change' would be advantageous before presentation of Figure 2 and its associated data in the paper.

1.4. We have elaborated the explanation of the term "relative risk" in the Methods (revised section 3.3.2) and at the beginning of the relevant Results section (revised section 2.2): "The results - 'effect estimates' - are reported as percent change in relative risk (RR) compared to a scenario in which all other variables are held constant."

It might be easier to interpret the overall results if Figures 2 and 4 were combined (as has been done in the explanatory text) so that there are 6 sub-figures within one overall figure.

Again the section lines 203-218 cross-refer to Figures 2 and 4 and would be better placed nearer that section with all the sections of self-reported health grouped together subsequently. As the authors note, there were relatively few responses to the self-reported symptoms questionnaire and these responses are likely to be prone to bias.

1.5. We have significantly revised the results section in order to make it flow better and remade the figure, combining Figures 2 and 4 (revised Figure 2).

The lower effect seen on non-advisory mature plume days compared with advisory mature plume days is important as it implies that the fact of an advisory being issued may in itself change health behaviours among the population, as discussed in line 248.

1.6. We agree with this comment and we have extended the discussion of this important finding (revised section 2.4): “The results indicate that public advisories changed health behaviours among the population. An increase in AMD associated with issuing of public advisories suggests that patients were taking precautionary measures. On the other hand, where mature plume days were not included in a public advisory, there are increased outcomes for HED visits, and to some degree PCMD, particularly in the elderly age group. This suggests that public advisories were useful for protecting vulnerable individuals, and that HED and PCMD may be truer proxies for acute health care needs than AMD.”

We also point out that the higher health effects associated with the mature plume compared to the primitive plume may at least partially be explained by the fact that the primitive plume was forecast and forewarned, while the mature plume was not (revised section 2.3).

Minor comments:

line 145 - can the units be specified i.e. assume the daily mean is the number of asthma medications dispensed, is this the number of people receiving medication or the number of items dispensed i.e. potential for a single individual to receive more than one item

1.7. We have added text to indicate more precisely that it is the number of individuals that are the outcome, so no one can be counted twice in one day (revised section 2.2): “We considered the impact of the mature plume on the daily number of asthma medication dispensing (AMD, daily-mean number of individuals 131 ± 70), MD consultations for respiratory disease in primary care centres (PCMD, 142 ± 63), and hospital emergency department visits (HED, 18 ± 8) for respiratory disease (Figure 2a-c) using regression analysis.”

And additionally in the Methods section 3.2 “As the same bout of illness is likely to result in recurring contacts with the health care system, we included only the first record of an individual’s health care contacts within a 14-day period for the same diagnosis category to avoid exposure misclassification with respect to the timing of the outcome.”

throughout the paper there are a number of typos e.g.

line 39: should be 'were' not 'was'

line 46: some words missing before 'qualitative'?

line 114: 'dispensing' not 'dispending'

The paper needs careful proofreading.

1.8. Thank you for pointing out these issues, they have been corrected

2. Reviewer #2

The findings of the study indicate an association between the exposure to mature (sulphate rich) volcanic plume and acute health effects in the exposed population, which has implications for the volcanic monitoring strategies, health risk assessment and population safety management in volcanic crises, globally. I trust the study is relevant for and should be of interest to, a wider scientific community (volcanologists, public health) and public (organisations) alike.

The study resembles some of the latest publications by the authors (e.g., Carlsen et al. 2019, under review at medRxiv) where the same approach has been used to correlate the air quality data with health care utilisation data for the same eruption (and study period). These data are unique in their high temporal resolution and their value has been well recognized, and acknowledged, in this paper; the study is justified and builds on the earlier work by focusing on specific events (mature plume days) that deserved more attention and in-depth analysis. Although the conclusions on the specific effects of mature plume remain somewhat reserved, the results demonstrate the differences in the recorded effects in days of the primitive and mature plume and thus show the need for consideration of their distinctive features and effects in the future studies. I consider the presented information important for advancing our understanding of the complexity of volcanic emissions and associated health effects, and specifically to serve for informing protection in volcanic areas that are not (well) monitored and lack robust data. Considering this, and that I see no major issues with the representation or discussion of the data, I kindly recommend this paper for publication.

The paper is overall well written, with a good balance of information and length. The statistical analyses seem sound and the methodology well-reasoned and sufficiently detailed, though, admittedly, I am no expert in such analyses. I added some (minor) comments/suggestions directly in the manuscript file, as I felt that additional clarification is necessary for certain parts of the text.

Thanks very much to the Authors and Editors for your understanding and patience to receive this review in these uneasy times.

Very best wishes,

Dr. Ines Tomašek

2.1. We thank Dr Tomašek for the thorough review and multiple, detailed comments on the manuscript file which have improved the manuscripts.

In particular, we have clarified the terminology related to mature/primitive plume types and sulphate/sulphuric acid/acid sulphate, and made it consistent throughout.

We have also added additional discussion about threshold recommendations; and additional references & review of previous studies where this was suggested. From revised section 2.7 “We recommend that for volcanic events which impact air quality at ground level, public advisories for both primitive and mature plumes are issued to protect the vulnerable groups. Our observational study cannot demonstrate whether sulphate was the causal agent for the observed health impacts, or simply a marker for one or more toxic constituents in the mature volcanic plume. However, sulphate formed through SO₂ gas-to-PM conversion is relatively simple to include in plume dispersion simulations and could be used as a proxy for mature plume advection into populated areas. The health impacts observed by us were associated with exposure to a plume with relatively low sulphate concentrations (3-8 µg/m³). In the absence of evidence for a ‘safe’ concentration of sulphate and/or other PM components in the mature plume, the public advisories should be based on a simple ‘plume present’

message using a single threshold. The threshold of 5 $\mu\text{g}/\text{m}^3$ of sulphate suggested previously by Lioy and Waldman²⁸ is too high for this purpose, and a lower value ($<1 \mu\text{g}/\text{m}^3$ above the non-volcanic background) would be more appropriate. Our recommendations are in line with those of ‘Review of evidence on health aspects of air pollution project’ by World Health Organisation³⁵, and would enable the public to take necessary steps to avoid exposure or mitigate the risk.”

We do not detail responses to minor comments, such as typos, or requests for minor clarifications – but we have amended these as can be seen in the revised manuscript file.

3. Reviewer #3

The authors examine the hypothesis that volcanic sulfur dioxide gas and sulfate aerosols are associated with increases in dispensing of anti-asthma medications, primary care visits and emergency department visits for respiratory complaints. The authors focus in particular on the effects of mature plumes of volcanic air pollution, as these sulfate-rich plumes are less well studied and, because they can be low in the criterion gas SO₂, may not be detected as a plume of volcanic air pollution to trigger an advisory.

The authors conclude that these sulfate-rich plumes are associated with increased anti-asthma medication dispensing, visits to primary care clinics and emergency departments for respiratory complaints.

The findings are not novel and are incremental to another preprint by the authors (<https://www.medrxiv.org/content/10.1101/19013474v1.full.pdf>) that more satisfactorily presents the methodology and details of the demographics and health record inquiry. Compared to the more fundamental finding of that preprint (that there was excess utilization for respiratory complaints during the eruption, compared to pre-eruption) this manuscript adds smaller insights about the relative lags in anti-asthma medications, clinic and ED visits.

3.1. We thank the reviewer for their comments. We regret that the novelty of this study was not sufficiently well explained and we are grateful for the opportunity to clarify it.

State-of-the-art novelty: Very little is known about health impacts associated with mature, PM-rich volcanic clouds. This is especially true in comparison to impacts of sulphur dioxide (SO₂), the most reliably and commonly monitored air pollutant in ash-poor eruptions. For SO₂, there are evidence-based exposure thresholds for different concentration levels, which are used when issuing public advisories (e.g. Barsotti et al. 2020 <https://doi.org/10.1016/j.jvolgeores.2019.106753>). Therefore, we politely disagree that our findings are only “incremental” to our previous publication which focussed solely on SO₂. We would even argue that it was not a surprising result that acute SO₂ exposure was associated with negative health impacts. On the other hand, it was an unexpected and novel finding that exposure to a low-SO₂, high-sulphate PM cloud had as significant impacts as we were able to demonstrate in this study.

Furthermore, we show that the observed increases in health impacts cannot be solely attributed to an increase in PM_{2.5} (see discussion in revised section 2.6). This suggests that the chemistry of the PM in the mature plume may be a causal factor. This is an important result, as evidence base is still weak on whether chemical components of the PM mass are individually responsible for observed negative health outcomes.

Impact novelty: We highlight that the air pollution episodes caused by the mature plume were not considered in the real-time air quality monitoring or the public advisories of air pollution in Iceland. This is because there was not a sufficient knowledge base to recognise that the associated health impacts are significant enough to warrant operational response. Our study shows this for the first time – the health impacts from this type of pollution are significant enough to warrant inclusion in eruption response planning. For example, we showed that the absence of public advisories was associated with increased health care usage (primary care and hospital emergency visits) for respiratory disease in vulnerable groups (revised section 2.4). We make more concrete recommendations for how this hazard may be included in operational eruption response (revised section 2.7)

Methodological novelty: this is the first time that a public health analysis of this size and quality is undertaken to examine the effects of a mature volcanic cloud. This makes our results more robust compared to previous studies on impacts of volcanic plumes of mature composition (e.g. vog studies from Hawaii, which are very important contributions but based on significantly smaller and less statistically strong databases). We now highlight this in the Introduction “To our knowledge, this study provides the most robust evidence to-date that ground-level exposure to a chemically mature volcanic plume leads to increased health care utilisation” and the Methods “This was the first time a population of this size and density (500 per km²) was assessed for health impacts of volcanic PM pollution using a population-wide register”.

It could add much more to our understanding by discerning which symptoms or conditions are exacerbated by sulfate-rich aerosols, which may penetrate deeper into lower airways than hygroscopic SO₂ (wheeze and sputum production vs nasal congestion and eye irritation), or if there are susceptible persons, such as those with chronic obstructive lung disease, heart failure, smoking history.

3.2. We agree with this comment. We have done additional analysis within the constraints posed by the available detail on covariates in the health care register. Information on most of the covariates mentioned by the reviewer (e.g. smoking status) was not available.

We compared primitive and mature plumes association within respiratory diagnosis subcategories (revised section 2.3) and were able to show that asthma medication dispensing and respiratory infections are higher in association with mature plume exposure. Other respiratory subcategories, including COPD, were largely non-significant due to a low number of events. We also analysed several non-respiratory diagnoses that were of interest and present the results in revised section 2.5. We observed a significant increase for circulatory system disease in children and adolescents. In HED, the risk of nausea and vomiting diagnoses at lag 0-1 and lag 0-2 was significantly increased in children and adolescents, but again, the number of events were very low.

Thus, the authors can provide more detailed methodology (or reference <https://www.medrxiv.org/content/10.1101/19013474v1.full.pdf>) and address additional hypotheses given the availability of the register.

3.3. We have added all relevant details about the methods in revised Methods sections 3.2 and 3.3. Code which is used in the analysis in this study (and not in the preprint) is presented in the supplement.

(Compared to the health registry data, the self report questionnaire adds little substantial information.)

3.4. We agree with the reviewer that the self-report questionnaire is less robust than the health register. However, we think that the self-report results add to the story and deserve to be mentioned as long as the caveats are made clear. We have phrased it in the following manner (revised section 2.5): “It is important to note that self-reporting by self-selected participants is prone to bias. However, the results may suggest that the health impacts of volcanogenic air pollution were potentially more varied than identified in the register-based analysis. This is potentially due to the fact that people are relatively unlikely to present to paid healthcare services when experiencing mild or transient symptoms”

There are typographical errors, such as:

Line 39: was-> were

Line 56 low levels Less than 200 ppb?

Line 57 airways “rei”?

and more.

3.5. Thank you for pointing this out. The manuscript has been carefully proof-read

In Methods, this site is no longer active?
<http://vedur.maps.arcgis.com/apps/OnePane/azuretime/index.html?appid=da54a233cb0d4b679bdf5a1a860d11b9> Shows no responses

3.6. We checked the link on Google Chrome Internet Explorer and Mozilla Firefox and found it to work

REVIEWERS' COMMENTS

Reviewer #1 (Remarks to the Author):

Thank you for the opportunity to re-review this paper. I found this version to be greatly improved from the previous one, especially in the readability of the results and of the flow through of the key findings. These were all much less confused than before. The paper also more clearly distinguished between the main findings on the health effects of the primitive and mature plumes, and downplayed the less reliable results n self-reported health. The authors response to the previous comments was comprehensive and I am satisfied that all of my concerns have been addressed in this new version.

I did not systematically proof-read the paper, but there seem fewer errors than previously. However, I did notice a few small corrections needed:

Line 251 - repetition of response in the sentence. needs rephrased ("response of the immune system response")

Line 280 - I think the 2nd 0-1 should be 0-2?

Suppl figure 3(e): the x-axis tick point labelling needs edited to be consistent with the other plots (age group categories)

Section 3.2 2nd line: there is an extra)

Reviewer #2 (Remarks to the Author):

I thank the authors for considering the suggestions and making the revisions to the paper. The authors addressed all my 'concerns' and I find the paper is much improved overall; I look forward to see it published.

One minor correction is needed for the reference in the lines 518-519: the findings you refer to are from the study Tomašek, I., Horwell, C.J., Damby, D.E. et al. Combined exposure of diesel exhaust particles and respirable Soufrière Hills volcanic ash causes a (pro-)inflammatory response in an in vitro multicellular epithelial tissue barrier model. Part Fibre Toxicol 13, 67 (2016).

<https://doi.org/10.1186/s12989-016-0178-9>, so the reference [59] should be adapted.

Best,
IT

Reviewer #3 (Remarks to the Author):

I find the rebuttal, revisions, inclusion of supplemental information highly responsive to my concerns and recommend publication.

Point by point response to REVIEWERS' COMMENTS

Reviewer #1 (Remarks to the Author):

Thank you for the opportunity to re-review this paper. I found this version to be greatly improved from the previous one, especially in the readability of the results and of the flow through of the key findings. These were all much less confused than before. The paper also more clearly distinguished between the main findings on the health effects of the primitive and mature plumes, and downplayed the less reliable results n self-reported health. The authors response to the previous comments was comprehensive and I am satisfied that all of my concerns have been addressed in this new version.

>> We thank the reviewer

I did not systematically proof-read the paper, but there seem fewer errors than previously. However, I did notice a few small corrections needed:

Line 251 - repetition of response in the sentence. needs rephrased ("response of the immune system response")

>> Done

Line 280 - I think the 2nd 0-1 should be 0-2?

>> Yes, corrected

Suppl figure 3(e): the x-axis tick point labelling needs edited to be consistent with the other plots (age group categories)

>>Done. Please note that this is now supplementary figure 2

Section 3.2 2nd line: there is an extra)

>> Corrected

Reviewer #2 (Remarks to the Author):

I thank the authors for considering the suggestions and making the revisions to the paper. The authors addressed all my 'concerns' and I find the paper is much improved overall; I look forward to see it published.

One minor correction is needed for the reference in the lines 518-519: the findings you refer to are from the study Tomašek, I., Horwell, C.J., Damby, D.E. et al. Combined exposure of diesel exhaust particles and respirable Soufrière Hills volcanic ash causes a (pro-)inflammatory response in an in vitro multicellular epithelial tissue barrier model. Part Fibre Toxicol 13, 67 (2016).

<https://doi.org/10.1186/s12989-016-0178-9>, so the reference [59] should be adapted.

Best,

IT

>> We thank the reviewer and have amended the reference

Reviewer #3 (Remarks to the Author):

I find the rebuttal, revisions, inclusion of supplemental information highly responsive to my concerns and recommend publication.

>> We thank the reviewer